# Progesterone and Inflammatory Response in the Oviduct during Physiological and Pathological Conditions

**DOI:** 10.3390/cells11071075

**Published:** 2022-03-23

**Authors:** Emily A. McGlade, Akio Miyamoto, Wipawee Winuthayanon

**Affiliations:** 1Center for Reproductive Biology, School of Molecular Biosciences, College of Veterinary Medicine, Washington State University, Pullman, WA 99164, USA; emily.harris3@wsu.edu; 2Global Agromedicine Research Center (GAMRC), Obihiro University of Agriculture and Veterinary Medicine, Obihiro 080-8555, Japan; akiomiya@obihiro.ac.jp

**Keywords:** cytokines, epithelium, inflammation, interleukins, oviduct, pregnancy, progesterone, toll-like receptor

## Abstract

Progesterone has been shown to be a potent suppressor of several inflammatory pathways. During pregnancy, progesterone levels increase, allowing for normal pregnancy establishment and maintenance. The dysregulation of progesterone, as well as inflammation, leads to poor pregnancy outcomes. However, it is unclear how progesterone imbalance could impact inflammatory responses in the oviduct and subsequently result in early pregnancy loss. Therefore, in this review, we describe the role of progesterone signaling in regulating the inflammatory response, with a focus on the oviduct and pathological conditions in the Fallopian tubes.

## 1. Introduction

It is well established that progesterone (P_4_) is crucial for mammary gland development, ovarian function, and uterine function. Furthermore, it has been addressed that the inflammatory response plays a large role in maternal–fetal tolerance, as well as pregnancy establishment and maintenance, including embryo development, embryo implantation, and parturition. However, physiological actions of P_4_, as well as its role in regulating inflammatory responses in the oviduct, is less understood compared to other reproductive tissues. Therefore, in this review article, we aim to provide a comprehensive literature review on the current understanding surrounding the role of P_4_ in immunomodulatory responses in the oviduct, inflammatory conditions in the oviduct and their impact on pregnancy outcomes, and perspectives for relevant future research questions. 

## 2. Oviductal Function and Pregnancy Establishment

The oviduct, or Fallopian tube in humans, is the tubal structure of the upper female reproductive tract that connects the ovary to the uterus. The oviductal environment is crucial for sperm migration, egg pick-up/transport, fertilization, preimplantation embryo development, and embryo transport to the uterus, where the embryo will reside for the remainder of pregnancy. The oviduct is the first environment that embryos are exposed to and therefore heavily impacts pregnancy establishment, including successful fertilization, embryo quality and birth outcomes [1]. If the environment becomes hostile within the oviduct, it can lead to early embryo death and disrupt normal progression to the blastocyst stage [2]. 

The oviduct is comprised of four regions: the infundibulum, which is responsible for egg pick-up from the ovary following ovulation; the ampulla, where fertilization occurs; the isthmus, where the embryo undergoes several cleavage divisions; and the utero-tubal junction (UTJ), the segment connecting the oviduct to the uterus. The oviduct is comprised of multiple cell types, including stromal cells (fibroblasts), ciliated epithelial cells, secretory epithelial cells, endothelial cells, and smooth muscle cells. These cells are tightly regulated by ovarian steroid hormones [3]. Oviductal epithelial cells directly interact with sperm [4], eggs [5], and embryos [6]. Furthermore, co-culturing gametes and embryos with oviduct epithelial cells promotes oocyte maturation [7], prolongs sperm motility [8], and improves fertilization rate [9] and embryo quality [10] in several mammalian species. These findings highlight that gamete/embryo interactions with the oviduct epithelium are crucial for pregnancy establishment. A previous in vivo study showed that the dysregulation of estrogen (E_2_) signaling in the oviductal epithelial cells alters the inflammatory response and ultimately leads to preimplantation embryo death in mice [2]. However, the roles of P_4_ in the immune/inflammatory regulation of the oviduct are unclear. Therefore, in this review, we summarize past and current findings on P_4_ and its action on oviductal immune responses and female reproductive function, especially during the establishment of pregnancy.

## 3. Progesterone Action

P_4_, the pregnancy hormone, is mainly secreted by luteal cells of the corpus luteum in the ovary following ovulation and is secreted by trophoblast cells during pregnancy [11]. Classical P_4_ signaling acts through the genomic signaling of nuclear progesterone receptors, PGR-A and PGR-B. Non-classical P_4_ signaling acts though non-genomic signaling of membrane receptors including G-protein-coupled membrane progestin receptors (mPRs) and P_4_ receptor membrane components (PGRMCs). P_4_ can also signal through nuclear glucocorticoid receptors (nGR) [12]. P_4_ regulates epithelial cell morphology and function in the oviduct to allow for normal pregnancy establishment. In the oviduct, increased circulating levels of P_4_ lead to de-ciliation in both non-human and human primates [13,14]. The treatment of P_4_ decreases ciliary beat frequency (CBF) in mice [15], guinea pigs [16], and humans [17], resulting in an overall deceleration of oocyte/embryo transport in the oviduct. This ‘P_4_-induced decreased CBF’ effect appears to be regulated through a classical PGR-dependent mechanism [16,18], partly by a modulation of intracellular Ca^2+^ influx through the transient receptor potential vanilloid (TRPV) 4 channel [19]. Indeed, a global loss of PGR in *Pgr*^−/−^ mouse models shows that genes involved in oocyte/egg transport are altered in the *Pgr*^−/−^ oviduct [20]. In addition to its effect on CBF, P_4_ also decreases muscle contractions in human Fallopian tubes [21]. Moreover, increasing levels of P_4_ from the follicular fluid after ovulation allows for sperm capacitation in hamsters [22]. In contrast to P_4_, E_2_ has opposed effects on epithelial cell ciliation, CBF, and muscle contractility in the oviduct (review in [23]). For the specific action of P_4_ in other reproductive tissues, refer to other review articles in this Special Issue. 

P_4_ has been shown to play a role in suppressing inflammation during pregnancy and within oviduct epithelial cells. However, it is not well understood how P_4_ regulates the inflammatory response in the oviduct to allow for preimplantation embryo development and pregnancy establishment. Therefore, in this review, we focus on P_4_ action on immune function in the oviduct. As such, we also discuss inflammatory pathways, immune responses, inflammation in the reproductive tract, and how P_4_ might affect both immune and reproductive functions and ultimately determine pregnancy outcomes.

## 4. Immunomodulatory Responses and Regulatory Pathways

Immune cells of the innate and adaptive immune response within the oviduct are responsible for maintaining an environment that supports oviductal cell/tissue integrity, pregnancy establishment, and ridding the oviduct of infection. As there are multiple players and regulatory pathways in the immune response, we have included a brief overview of innate and adaptive immunity (Figure 1) before describing their actions in the oviduct and the reproductive tract. Innate immunity refers to immune responses that block the entrance of a foreign agent, such as a physical barrier, or the recruitment of immune cells (such as macrophages and neutrophils) to kill and remove foreign antigens. Within the oviduct, the epithelial layer and mucosa also act as a physical barrier against foreign antigens, including bacteria, viruses, and parasites. Adaptive immunity refers to immune responses that are antigen-specific and create antibodies against the pathogen for future contact. 

### 4.1. Innate Immune System 

In general, the molecular signaling of innate immunity involves the activation of toll-like receptors (TLRs) [25] and, subsequently, the recruitment of the TLR-domain containing adapter proteins, including MyD88/IRAK1/IRAK4 (myeloid differentiation primary response 88/interleukin receptor-associated kinase 1/4). This protein complex then activates TRAF6 (TNF receptor associated factor 6), which results in the activation of IKKα/β (inhibitor of κ-B kinase α and β). Active IKKα/β then phosphorylates IκBα (IκB kinase α), leading to IκBα degradation and the nuclear translation of NFκB (nuclear factor kappa B) translocation into the nucleus. The activation of NFκB, a transcription factor, leads to cytokine production. Cytokines often include interleukins (ILs), chemokines, CXC-chemokine ligands (CXCLs), interferons (IFNs), granulocyte macrophage colony-stimulating factor (GM-CSF), tumor necrosis factors (TNFs), and others. These cytokines often act as a chemoattractant and provide communication between immune and non-immune cells that govern both innate and immune regulatory pathways.

In addition to systemic innate immune responses, physical barriers (such as tight junctions, epithelial/mucus membranes, and mucus) also provide innate immunity. These physical barriers act as protective layers against pathogen invasion. The epithelial layer expresses cell receptors that can detect the presence of pathogens and signal defensive responses (such as TLRs). These epithelial cells can also produce cytokines in response to both microbials and non-microbial stimulants, such as hormones and stress. Moreover, epithelial cells can secrete anti-microbial peptides (AMPs) in response to both hormones and pathogens. These peptides include defensins (DEFs), lactoferrin, cathelicidins, the secretory leukocyte protease inhibitor (SLPI), and others [26]. As such, epithelial cells provide crosstalk to subsequently stimulate adaptive immune responses [27]. Therefore, epithelial cells serve as mediators and regulators for both innate and adaptive immune responses. The epithelial-mediated immune response has been termed ‘*epimmunome*’ [28], and is crucial for the first defense mechanism and homeostasis within tissues.

### 4.2. Adaptive Immune System 

To activate adaptive immunity, antigen presenting cells recognize the foreign antigen, break it apart, and then present a fragmented antigen to a naïve T cell. As such, naïve T (Th0) cells behave similarly to a stem cell, as they can differentiate into Th1, Th2, Th17, or Treg depending on which cytokines are present. Th1 and Th17 cytokines induce pro-inflammatory responses, whereas Th2 cytokines induce anti-inflammatory responses. Ideally, Th1 and Th2 pathways often work in concert to regulate inflammation. However, if either Th1/Th17 or Th2 become dominant, this creates an imbalance leading to excessive inflammatory responses. Inflammatory/anti-inflammatory ratios may present as Th1/Th2 or Th17/Th2. Th1 responses produce long-lasting immunity, including adaptive immunity, inflammation, cytotoxicity, and delayed-type hypersensitivity [29]. In contrast, Th2 responses produce short-acting immunity (first responder), such as innate immunity, and decrease inflammatory Th1 cells [29,30]. 

Cell-mediated immunity does not rely on antibodies, but instead relies on T cells, macrophages, and cytokines. Cell-mediated immunity provides protection against intracellular pathogens. When a foreign antigen is present, major histocompatibility complex (MHC) class I proteins present the antigen on the cell surface. The antigen is then recognized by T lymphocytes. When a helper T cell recognizes the antigen on MHC class I proteins, it signals for the release of cytokines. Cytokines produced by helper T cells then recruit killer T cells and macrophages to destroy the antigen presenting cells. For more detailed information regarding specific cytokines/chemokines and their receptors, see review [29].

Although it has been suggested that P_4_ suppresses inflammation [31,32], there is a limited body of literature regarding how P_4_ modulates these innate and adaptive immune responses in the oviduct in the context of pregnancy establishment. Therefore, we have provided a review in the following sections on the action of P_4_ on the regulation of innate vs. adaptive immune response in general, and in the oviduct, whenever applicable. 

## 5. Immune Cells and Epithelial Immunity in Human Fallopian Tubes 

The Fallopian tube was originally thought to be a sterile environment. However, recent studies show that there are resident immune cells in the Fallopian tube even in the absence of infection [3,33,34,35,36,37]. Immunohistochemical, fluorescent-activated cell sorting (FACS), and single-cell RNA-sequencing (scRNA-seq) analyses reveal that immune cells in both innate and adaptive immune responses are present in human Fallopian tubes [3,33,34,35,36,37]. Leukocytes (expressing PTPRC, also known as CD45) are the most abundant in human Fallopian tubes compared to other reproductive tissues. We have re-analyzed previously published scRNA-seq data from human Fallopian tubes [3] and found the presence of *PTPRC*+ cells (Figure 2) that are consistent with previous reports [37]. T cell marker *CD52*+ has also been detected at a high abundancy. However, *CD14*+ monocytes/macrophages [3] and *NCAM1*+ (also known as *CD56*+) NK cells are present at low levels compared to other immune cell types [3,35]. Recent scRNA-seq data from pooled human Fallopian tubes show that tissue *CD69*+ resident memory T cells are also detected [37]. However, B cell maker *CD19* is not detectable in these two scRNA-seq datasets [3,37]. Interestingly, Hu et al. found that these *CD69*+ cells also express an epithelial cell marker, *EPCAM*+. This suggests that T cells and other innate immune cell types reside in human Fallopian tube tissues and can modulate inflammatory responses in the presence of foreign bodies such as embryos (semi-foreign), sperm, or pathogens. 

In addition to systemic innate and adaptive immune cells, we found that TLRs, such as *TLR5* (Figure 2) and *TLR3* (data not shown), are also expressed in *EPCAM*+ epithelial cells in human Fallopian tubes. Defensins (DEFs) are secreted from secretory epithelial cells in female reproductive tissues [38]. Here, we showed that β-defensin (*DEFB1*) is also detected in Fallopian tube epithelial cells (Figure 2). Similarly, SLPI is also present in the *EPCAM*+ cell population (Figure 2), consistent with previous findings that SLPI protein is expressed in the epithelial cells of human Fallopian tubes [39]. Together, these data indicate that the human Fallopian tube is an immunomodulatory dynamic tissue.

## 6. Progesterone Action on Immune Responses 

As mentioned earlier, there has been limited research on the action of P_4_ on the inflammatory response specifically in the oviduct. As such, we provide information regarding P_4_ on immune responses in reproductive and other tissues in this review, and in the oviduct when the information is available.

### 6.1. P_4_ and Innate Immune System 

In the reproductive tissues including the uterus and cervix, P_4_ suppresses E_2_-induced pro-inflammatory responses in a PGR-dependent manner [40] while promoting anti-inflammatory responses [41,42]. For its action in innate immunity, P_4_ can both suppress and activate macrophages depending on the tissue type (review in [43]). In human endometrium, macrophage infiltration is increased before menstruation [44], which coincides with elevated P_4_ levels. In addition, P_4_ increases the production of prostaglandin E_2_ (PGE_2_) by macrophages, which inhibits the production of GM-CSF by uterine epithelial cells [45]. Using a cell impermeable form of P_4_ to activate mPRs, Lu et al. showed that P_4_ increases *Il1b*, *Tnf*, prostaglandin synthase 2 (*Ptgs2*), and nitric oxide synthase 2 (*Nos2*) transcripts in the mouse macrophage (RAW264.7) cell line [46]. These studies suggest that P_4_ can directly impact the function of macrophages. Moreover, mPRs, unlike nuclear PGR, are detected in macrophages [43,47]. Therefore, P_4_-dependent actions in macrophages may be mainly modulated by mPRs as opposed to classical nuclear PGR signaling. The withdrawal of P_4_ increases expressions of *IL8* and monocyte chemoattractant protein-1 (*MCP1*) transcripts in human endometrial explants [48], suggesting that P_4_ suppresses these cytokines in the uterine tissues. However, production of cytokines can be regulated by other cell types in the uterus, such as epithelial cells [49]. Therefore, it is still difficult to conclude whether P_4_ directly governs the production of these cytokines through immune cells or other cell types in the reproductive tissues, as the cytokines were not directly extracted from the immune cells themselves.

### 6.2. P_4_ and Adaptive Immune System 

Peripheral blood mononuclear cells (PBMCs) are often used to study both innate and adaptive immune responses as PBMCs originate from hematopoietic stem cells and can differentiate into dendritic cells, monocytes, and lymphocytes. Ndiaye et al. isolated T cells, B cells and monocytes from PBMCs and found that progestin and adipoQ receptor family members (PAQRs, also known as mPRs) are present in T cells, but not in B cells or monocytes [50]. More specifically, *PAQR7* (mPRα) and *PAQR5* (mPRγ) transcripts are highly expressed in CD4+ T cells compared to γδ+ and CD8+ T cells [50]. Moreover, using immunofluorescent analysis, mPRα protein was detected on the cell membrane of T lymphocytes (CD4+ T cells) isolated from bovine corpora lutea (CL), and [^3^H]-P_4_ was found to bind to the plasma membrane of T lymphocytes [50]. These findings indicate that P_4_ regulates the adaptive immune response, especially in T lymphocytes, in bovine luteal tissues.

Dendritic cells serve as a bridge between innate and adaptive immune responses as they are responsible for initiating adaptive antigen-specific T cell responses. Butts et al. assessed the differences in bone marrow-derived dendritic cell (BMDC) concentration and response to steroid hormones in male and female rats [51]. It was shown that immature BMDCs isolated from females expressed PGR protein at higher levels compared to male rats [51]. 

As lipopolysaccharide (LPS), an endotoxin from bacteria, is generally used to induce pro-inflammatory responses, it has previously been shown that LPS increases NFκB, TNFα, and IL1β production in the oviduct [52,53]. P_4_ treatment decreases (LPS)-induced pro-inflammatory cytokines including TNFα and IL1β, and the effect of P_4_ is abrogated by the co-treatment of PGR antagonist (RU486) [51]. However, P_4_ treatment has no effect on LPS-induced IL10 production in BMDCs [51,54]. Although these cytokines are expressed at lower levels in females compared to males, the impact of P_4_ on decreasing LPS-induced cytokines is much more significant in females compared to male rats. It is likely that this phenomenon is due to the fact that BMDCs isolated from females express more PGR compared to male rats, indicating a sex-different response in PGR expression and immune responses in rat BMDCs. P_4_ also decreases LPS-induced dendritic cell-associated markers (MHC class II and CD80), and inhibits T lymphocyte proliferation in female rats [54]. In human endometrium, a decrease in P_4_ levels results in an increase in IL8 (pro-inflammatory) [55]. These studies suggest that P_4_ has inhibitory effects on inflammation and may contribute to sex differences concerning the inflammatory response observed between men and women, especially after puberty [51,56,57,58]. In addition to PGR, Jones et al. found that P_4_ suppressed TLR3/TLR4-mediated pro-inflammatory cytokine production by LPS- or poly I:C (immunostimulant)-induced BMDCs through the glucocorticoid receptor (GR) [59]. In addition, P_4_ prevents TLR3/TLR4-induced IL12 production through both PGR and GR-mediated pathways [59]. These data suggest that the effect of P_4_ on TLR3/TLR4-mediated signals could be governed in both PGR- and GR-dependent manners. Therefore, in general, P_4_ stimulates anti-inflammatory responses while suppressing pro-inflammatory responses (Figure 3).

## 7. Progesterone Regulation of Inflammation during Pregnancy

It has been established that female steroid hormones (E_2_ and P_4_) modulate immune regulation in the female reproductive tract [33,60]. P_4_ has also been shown to overall inhibit inflammation by promoting the production of growth factors that subsequently disrupt cytokine activity and increase cellular repair pathways [32]. P_4_ circulatory levels are high during pregnancy; therefore, P_4_ is likely suppressing the inflammatory response to support maternal tolerance of the embryo/fetus. It was previously shown that lymphocytes from pregnant females produce cytokines in the presence of P_4_, and these cytokines are called P_4_-induced immunomodulatory proteins (PIBF) [61]. Szekeres-Bartho and Wegmann showed that PIBF increases IL10 (7-fold) in activated T cells isolated from mouse spleen cells [61]. This is crucial as IL10 prevents cytokine production by Th1-lymphocytes as well as CD8+ T cells. Therefore, a seven-fold increase in IL10 has beneficial anti-inflammatory properties during pregnancy [62]. These findings indicate that PIBF may act as one of the factors at the fetal–maternal interface to locally suppress the immune response. As there is limited research in the literature regarding the roles of P_4_ in the immune function of the oviduct, we also include findings in relation to P_4_, immune response, and pregnancy in this review. This is mainly because P_4_ action in other female reproductive tissues might also apply to the oviductal response. As such, we have included descriptions of several common cytokines, including whether these factors are pro- or anti-inflammatory and their response to P_4_ (Table 1). 

### 7.1. P_4_ and Its Effects on Immune Responses in the Oviduct

#### 7.1.1. Oviductal Cells

In addition to immune cells, mPRs and PGR are expressed in multiple cell types in mammalian oviducts, including the epithelium, fibroblasts, and muscle cells [77,78]. In human Fallopian tubes, Hess et al. show that genes in the immune and inflammatory pathways (such as *MCP1* (or *SCYA2*), *IL1β*, *IL8* (*CXCL8*), *CCL2*, *CCL18*, *CXCL2*, *CXCL3*, and TNFα-induced protein 2; *TNFAIP2*) are significantly downregulated in tissues collected at the luteal phase (high circulatory P_4_ levels) compared to the follicular phase (high E_2_ levels) (Figure 4) [79]. These data suggest that P_4_ acts to suppress immune and inflammatory cytokines after ovulation, at which time the eggs/embryos would be present in the Fallopian tube.

Kowsar et al. show that P_4_ suppresses LPS-induced transcriptional levels of *TLR2*, *TLR4*, *IL1β*, *TNFA*, *NFKBIA*, and *PTGS2* in bovine oviductal epithelial cell (BOEC) cultures [53], suggesting that P_4_ is crucial for the suppression of Th1-mediated immune responses to LPS (Figure 4) [53]. In addition, acute phase antimicrobial peptides, such as α1-acid glycoprotein (AGP), can be produced from epithelial cells to protect against pathogen invasion [80]. Kowsar et al. also show that *APG*, its receptor (*APGR*), and protein secretion of APG in BOECs are significantly induced after P_4_ treatment (Figure 4) [80]. Moreover, treatment of APG in BOECs suppresses *TLR2* and *TNFA* transcripts [80]. These findings indicate that P_4_ promotes the secretion of AMPs in the oviduct, such as APG, to inhibit pro-inflammatory cytokine production in bovine epithelial cells. However, these studies do not investigate whether the effect of P_4_ on cytokine production are mPRs- or PGR-dependent pathways. 

To gain insight on the regulation of P_4_ receptor signaling pathways on immune function of the oviduct, we assessed a study using the *Pgr*^−/−^ mouse model, which was previously shown to cause female infertility partly due to an ovulatory defect [81]. Akison et al. show that deletion of the classical *Pgr* leads to a predicted decrease in leukocyte infiltration compared to *Pgr*^+/−^ oviducts using ingenuity pathway analysis [20], suggesting that a loss of PGR may suppress leukocyte infiltration in the oviduct. However, there were no other cytokines or immune regulatory pathways that were significantly disrupted in this dataset, and it was not identified which genes were involved in this leukocyte infiltration pathway. Therefore, additional investigation regarding the impact of P_4_ and its receptors on immune functions of the oviduct remain to be evaluated. 

A recent study using in vitro buffalo oviduct epithelial culture, showed that the inflammatory response, specifically neutrophil recruitment, changes across the estrous cycle as E_2_ and P_4_ levels oscillate [82]. Yousef et al. found that neutrophil numbers were lower in the isthmus of the ipsilateral oviduct and higher in the ampulla of the contralateral oviduct during estrus. During diestrus, however, neutrophil numbers were stable in the ampulla and isthmus. Using scanning electron microscopy, Yousef et al. also show that neutrophils are present in the oviduct epithelial cell layer, within the basal lamina and oviductal lumen, suggesting that oviduct epithelial cells, under the influence of E_2_ and P_4_, are important for the recruitment of neutrophils within the oviduct [82]. 

In summary, these studies show that E_2_ and P_4_ can influence the inflammatory response within oviduct epithelial cells, including neutrophil recruitment. These findings also demonstrate that inflammation is tightly regulated in various regions of the oviduct (ampulla vs isthmus) and hormone levels from follicular fluid may impact the inflammatory response in the oviduct, as the side of ovulation also impacts inflammation. However, it is unclear how much P_4_ is directly involved in the regulation of the oviductal epithelial cell response in a physiological condition, as the majority of cytokine production was measured or evaluated in an LPS-induced environment.

#### 7.1.2. Sperm 

The final destination of the sperm is the ampulla region of the oviduct, where sperm fertilize the egg. It is established that P_4_ from the follicular fluid acts as a chemoattractant agent for sperm to migrate from the uterus towards the oviduct [83]. P_4_ also activates CatSper on the sperm to stimulate capacitation [84]. Recent studies have shown that regulation of the inflammatory response may also be important for sperm transport in the female reproductive tract, yet it is unclear how P_4_ may modulate immune responses during sperm migration.

The presence of seminal fluid and sperm has been shown to trigger maternal immune tolerance in the female reproductive tract [85]. Yousef et al. also found that sperm bind to BOECs and stimulate the expression of *IL10* and *TGFB1* in BOECs [86], suggesting an induction of anti-inflammatory responses by sperm. However, the presence of sperm after natural mating decreases *IL10* in the UTJ region of rabbit oviducts [87]. Bromfield et al. show that the expression levels of *Csf2*, *Lif*, and *Il6* are similar between oviducts collected at the estrus stage or after mating with intact or vasectomized male mice [88]. This suggests that cytokine responses to the presence of sperm could be region- and species-specific. A recent study using a bovine oviductal epithelial explant shows that sperm binding significantly induces the expression of *TLR2* and *IL8* [89]. More importantly, sperm-induced proinflammatory cytokine responses in the bovine oviductal explant are inhibited by TLR1/2 antagonist (CU-CPT22) treatment [89]. Although these studies implicate that the presence of sperm may stimulate TLR-mediated responses in the oviduct to modulate immune tolerance in preparation for the presence of embryos, it has yet to be investigated whether P_4_ from follicular fluid is directly involved in sperm-induced cytokine production in the oviduct. 

#### 7.1.3. Embryos

Preimplantation embryo development and embryo genome activation take place while embryos are located in the oviduct. During this time, circulating P_4_ levels rise due to a functional CL after ovulation. As such, it is crucial to evaluate how P_4_ might impact the immune environment in the oviduct that subsequently affects the quality of the embryos. It was previously shown that decreased P_4_ levels reduce the number of blastocysts in mice [90]. However, it is not evident whether P_4_ directly impacts the embryos or indirectly impacts them through the regulation of the immune environment in the oviduct. Recent studies showed that proper E_2_ signaling in the oviduct is indispensable for preimplantation embryo development [2]. Loss of estrogen receptor α (ESR1) in oviductal epithelial cells in mice has been shown to increase the expression of *Il17*, *Il17rb*, *Cxcl17*, and other AMP genes involved in innate immune responses [2], resulting in embryo death prior to the two-cell stage. Unfortunately, little is known regarding the action of P_4_ on the oviductal immune environment and its impact on embryo development. Nevertheless, these recent findings indicate that the microenvironment in the oviduct, controlled by E_2_ and P_4_, allows for optimal levels of inflammatory responses and is therefore conducive for proper preimplantation embryo development and the successful establishment of early pregnancy. 

In summary, P_4_ both suppresses pro-inflammatory and stimulates anti-inflammatory responses in the oviduct. Therefore, we can speculate that the presence of P_4_ from the follicular fluid in the oviduct prevents excessive pro-inflammatory cytokine production from the oviductal epithelial cells. This reduction in pro-inflammatory responses allows the oviductal environment to tolerate the presence of foreign bodies such as sperm and seminal fluid and promote sperm viability in the upper reproductive tract. This tolerative immune response to the sperm could also prime the oviduct to allow for the newly fertilized embryos to develop. Evidence suggests that optimal E_2_ signaling in the oviduct ensures proper inflammatory balances while embryos are present. Therefore, it is possible that P_4_ from the functional CL during the preimplantation embryo developmental stage could also antagonize E_2_-induced pro-inflammation while promoting anti-inflammatory responses to allow for embryo survival in the oviduct.

## 8. Progesterone and Oviductal Inflammation in Pathological Conditions

Unfortunately, the rate of reproductive tract infection is high, with approximately one million people contracting a sexually transmitted infection (STI) each day, worldwide [91]. There are 376 million reported cases with infection of either chlamydia, gonorrhea, syphilis, or trichomoniasis, each year [91,92]. Furthermore, the rate of congenital syphilis in women has increased over 172% during 2014–2018 [93]. Previous studies have shown that women have a higher prevalence of STIs compared to men, including Herpes Simplex Virus 2 (HSV2) [94], and are more vulnerable to contracting STIs for several reasons. Biologically, the female reproductive tract is equipped with an innate mucosal barrier. The vagina is also likely to be exposed to higher volumes of sexual secretions from the ejaculate compared to the penis [95]. In contrast to this high prevalence, women are less likely to be diagnosed with STIs because 70–80% of STIs in women are asymptomatic [96,97,98]. If left untreated, STIs can continue to migrate to the upper female reproductive tract, including the Fallopian tubes, resulting in Pelvic Inflammatory Disease (PID) and can cause lasting effects on oviduct function, such as ectopic pregnancy (described below), infertility, or adverse effects on the resulting offspring. 

Wira and Fahey suggest that the window of vulnerability for infection in women is approximately 14–23 days post menses, or the luteal phase [96]. This is mainly due to the effect of elevated P_4_ during the luteal phase that prevents vaginal epithelial keratinization [97]. As such, the innate protective barrier is absent, making it easier for pathogens to invade through a bare vaginal epithelial cell layer [97]. Therefore, in the lower reproductive tract, such as the vagina, raised P_4_ levels would result in increased pathogen susceptibility through the regulation of the epithelial cell barrier. However, in the upper reproductive tract such as the Fallopian tubes, P_4_ would modulate both innate and adaptive immunity to allow for proper regulation of sperm survival and embryo development by suppressing pro-inflammatory responses. In this section, we will focus on how P_4_ impacts the pathological states of the Fallopian tube such as during chlamydia infection, hydrosalpinx, and ectopic pregnancy. 

### 8.1. Chlamydia Infection 

*Chlamydia trachomatis* is an intracellular, obligate bacterium with two forms: the elementary body (does not replicate, but is infectious), and the reticulate body (does replicate, but is non-infectious). Chlamydia infection is the most common cause of PID, which can lead to tubal damage, hydrosalpinges (fluid buildup and blockage in the Fallopian tube), ectopic pregnancy, and infertility [99]. Twenty percent of women with chlamydia infection develop PID, 3% suffer from infertility, and 2% experience poor pregnancy outcomes [98]. Previous studies have shown that women with chlamydia infection or previous exposure have higher rates of spontaneous abortion [100]. Because chlamydia infection and PID cause damage to the epithelial lining of the oviduct and loss of membrane polarity, this epithelial damage could disrupt the normal expression and patterning of ion channels and epithelial membrane transporters leading to the formation of hydrosalpinx and abnormal oviductal secretions [101]. 

Several studies have evaluated the effect of P_4_ on chlamydia infection in the uterus and the vaginal tissues. However, little information is available regarding the impact of P_4_ on chlamydia infection in the oviduct. Although many studies demonstrate that P_4_ inhibits certain inflammatory responses, Tuffrey et al. show that P_4_ treatment prior to chlamydia infection enhanced lesion progression in the mouse oviduct [102]. In this mouse study, it was proposed that the treatment of P_4_ disrupts the estrous cycle, thereby halting normal epithelial shedding and creating an environment conducive for chlamydia progression [102]. 

In the uterine environment, Kintner et al. found that P_4_ suppresses E_2_-induced production of the infectious form (EB) of chlamydia and reduces infection rates in the endometrial epithelial/stromal cell co-culturing system [103]. In addition, Amirshahi et al. demonstrate that the treatment of P_4_ in endometrial epithelial cells (ECC-1) causes an increase in the expression of genes related to amino acid metabolism and carbohydrate metabolism including the TCA cycle and glycolysis in Chlamydia [104]. In rats, P_4_ treatment also aids in the development of chlamydia infection [105], potentially through local immunosuppression [106] in the uterus and vagina. Lymph nodes of P_4_-treated, but not E_2_- nor E_2_+P_4_-treated, ovariectomized rats secreted higher levels of IL10 and IFNγ compared to vehicle controls [106]. It was concluded that P_4_ enhances susceptibility for chlamydia infection through disruption of the normal estrous cycle and through immunosuppression that allows for the proliferation of pathogens. These data suggest that P_4_ affects innate immunity in the epithelial cell layer of the female reproductive tract, which subsequently impacts host–pathogen interactions, and that P_4_ systemically suppresses immune responses, increasing susceptibility during chlamydia infection. However, these studies assessed the impact of P_4_ only in the uterus and vagina; the oviductal tissues were not investigated.

### 8.2. Hydrosalpinx 

Hydrosalpinx results from the buildup of Fallopian tube fluid (hydrosalpinges) that causes blockage of the Fallopian tube and has been linked to poor pregnancy outcomes. Women with hydrosalpinx have a 50% reduction in pregnancy rate compared to other tubal-related fertility issues [107]. Hydrosalpinx has also been shown to increase the risk of developing an ectopic pregnancy [108,109]. A few hypotheses have been proposed surrounding the impact of hydrosalpinx on poor pregnancy outcomes: (1) fluid from the inflamed Fallopian tube leaks into the uterine cavity causing a hostile environment for embryo implantation; (2) excessive volumes of fluid could mechanistically impair embryo attachment; and (3) oviductal fluid from hydrosalpinx patients may be cytotoxic to preimplantation embryos, and therefore prevent normal embryo progression to the blastocyst stage. Additionally, hydrosalpinx has also been shown to disrupt normal implantation for a number of other reasons including an increased concentration of immune factors and cytokines (such as LIF, tropho-uteronectin (TUN), and IL1) [110,111,112], and decreased blood flow [64]. 

In addition to the aforementioned hypotheses, misregulation of P_4_ signaling pathways have also been included in the ethology of hydrosalpinx conditions. Devoto and Pino found that the concentration of E_2_ receptors and cytosolic P_4_ receptors were decreased in 11 hydrosalpinx patients that underwent salpingectomy compared to normal Fallopian tubes [113]. The nuclear P_4_ receptor is present at lower levels in hydrosalpinx, but not significantly different from normal Fallopian tubes. Binding affinities (Kd) of E_2_ and P_4_ are not different between hydrosalpinx and normal Fallopian tubes (measured via specific radioimmunoassay). Therefore, hormonal imbalances in the Fallopian tubes of hydrosalpinx patients could be due to the reduction in hormone receptor levels. This study is limited, however, in that it only assessed histological sections of the ampulla region and did not specify the type of nuclear and cytosolic E_2_ and P_4_ receptors. Moreover, it is still unclear whether decreased levels of P_4_ receptors could be the cause or the result of hydrosalpinx conditions in these patients.

### 8.3. Ectopic Pregnancy 

Ectopic pregnancy occurs in ~1–2% of pregnancies [114]. Implantation of the embryo in the Fallopian tube (tubal ectopic pregnancy) accounts for 98% of ectopic pregnancies [115]. If left untreated, ectopic pregnancy can result in rupturing of the Fallopian tube, leading to internal bleeding. As a result, tubal ectopic pregnancy rupture is the leading cause of maternal death during the first trimester, accounting for 6% of all maternal deaths [116]. Inflammation in the oviduct has also been shown to increase the risk of developing tubal ectopic pregnancy [115]. Specifically, inflammation due to chlamydia infection, has been associated with increased rates of ectopic pregnancy [117]. Because cytokines are required for invasion of the embryo in the endometrium, the upregulation of cytokines in the oviduct may create an environment suitable for embryo implantation in the case of inflammation or infection, leading to ectopic pregnancy recognition. In addition to cytokine production, it is also possible that the disruption of normal morphology and function of oviductal cells, especially ciliated epithelial cells, due to an infection or prolonged inflammation, could also result in ectopic pregnancy. 

In addition to inflammation and infection, the alteration of circulating P_4_ levels has also been associated with ectopic pregnancy. Lower P_4_ levels have been identified in women with tubal ectopic pregnancies compared to women with normal pregnancies [118]. However, P_4_ is not sensitive enough to be used as a sole biomarker for ectopic pregnancy [119]. In contrast, high circulating levels of synthetic P_4_ from the use of levonorgestrel are also associated with ectopic pregnancy [120]. As mentioned above, changes in P_4_ concentrations can affect the function of epithelial cells of the oviduct. Therefore, these studies indicate that an optimal level of P_4_ is required for the normal function of the Fallopian tubes.

Additionally, levels of PGR should also be considered into the etiology of ectopic pregnancies. PGR is present in the Fallopian tubes from both ectopic and normal pregnancies. Marx et al. observed that PGR was highly expressed in ectopic pregnancy tubal epithelium [121]. However, Land et al. found that Fallopian tubes from ectopic pregnancy patients tended to have decreased PGR and ESR expression [122]. Therefore, there are no clear or stand-alone biomarkers for ectopic pregnancies, except for current human chorionic gonadotropin and ultrasound for embryo/fetus detection tests [123].

## 9. Perspectives and Concluding Remarks

P_4_ is a potent suppressor of the inflammatory response. It has multiple potential implications, such as offering a treatment for women with recurrent miscarriages, and a role in many female contraceptives that may increase susceptibility to STIs. Because there are many possible reasons for miscarriage, or recurrent miscarriage, it would be of use to utilize a screening panel of cytokine levels to determine proper treatments to reduce pro-inflammatory responses. Furthermore, there is still quite limited information available on the P_4_ regulation of cytokines in the context of the Fallopian tubes.

The impact of P_4_ on the human Fallopian tube after chlamydia infection should also be further evaluated, and the link between contraceptive use and infection rates should be more highly studied and discussed when prescribing female contraceptives. Because contraceptive use could impact susceptibility for chlamydia infection, it would be of use to implement regular screenings during yearly check-ups, especially since 70–80% of chlamydia-infected women are asymptomatic. This could reduce long-term damage and progression to PID, and other issues, such as ectopic pregnancy or infertility.

Many studies previously discussed in this review have suggested that pro- and anti-inflammatory cytokines are present in epithelial cells during infection and pregnancy, and are differentially regulated by E_2_ and P_4_. However, most studies looking at the hormonal regulation of inflammation have focused on the uterus, closely examining implantation and maternal–fetal tolerance. Few studies have assessed the hormonal regulation of inflammation in the oviduct and its impact on pregnancy establishment. Furthermore, the findings on the hormonal regulation of inflammation in the oviduct are mainly infection-centric, such as chlamydia infection, as opposed to the effects on preimplantation embryo development. 

To better understand the role of classical progesterone signaling within oviductal epithelial cells, PGR could be conditionally ablated from the mouse oviduct epithelium. This mouse model could be used to assess preimplantation embryo development, embryo transport, fertilization, sperm migration, and cytokine profiles to answer important questions related to the effect of inflammation on preimplantation embryo development, ectopic pregnancy, and pregnancy establishment in mammals.

## Figures and Tables

**Figure 1 cells-11-01075-f001:**
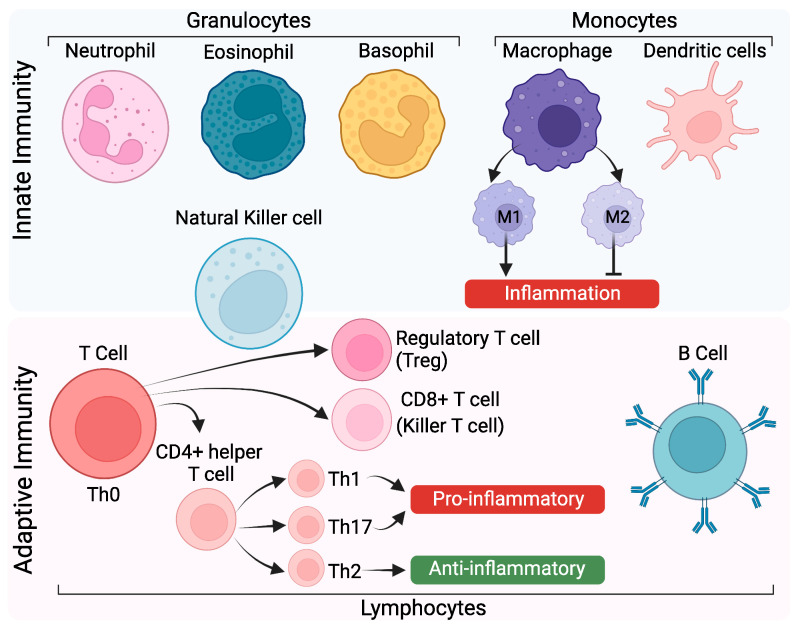
Innate and adaptive immune response. Innate immunity: There are 3 different types of leukocytes: granulocytes, monocytes, and lymphocytes. Granulocytes are classified into neutrophils, eosinophils (acidophiles), and basophils. Neutrophils are phagocytes and are the most abundant leukocytes that act as one of the first responders of the innate immune system. Eosinophils contain acidophilic cytoplasmic granules. Basophils are the least common granulocyte within the innate immune system. Monocytes give rise to macrophages and dendritic cells. There are 2 types of macrophages: M1 and M2 macrophages. Macrophages may also recruit lymphocytes for the adaptive immune response. M1 macrophages promote inflammation while M2 macrophages suppress inflammation and stimulate tissue repair [24]. Dendritic cells are antigen presenting cells for T cells and B cells that act as messengers between the innate and adaptive immune system. Natural killer (NK) cells are produced from lymphoid progenitors but act as part of the innate immune response. Adaptive Immunity: Lymphocytes include T cells, B cells, and NK cells. There are 3 types of T cells: (1) CD4+ helper T cells that release cytokines to signal other immune cells of the presence of infected cells, (2) CD8+ killer T cells that are cytotoxic and directly kill infected cells, and (3) regulatory T cells that act as general suppressors of the inflammatory response. Naïve T cells differentiate into either helper or killer T cells, while B cells produce antibodies in response to pathogenic antigens for the adaptive immune response. Adaptive immunity also includes T-helper 1 (Th1; cell-mediated), Th2 (humoral), Th17, and T regulatory (Treg) responses. Created with BioRender.com.

**Figure 2 cells-11-01075-f002:**
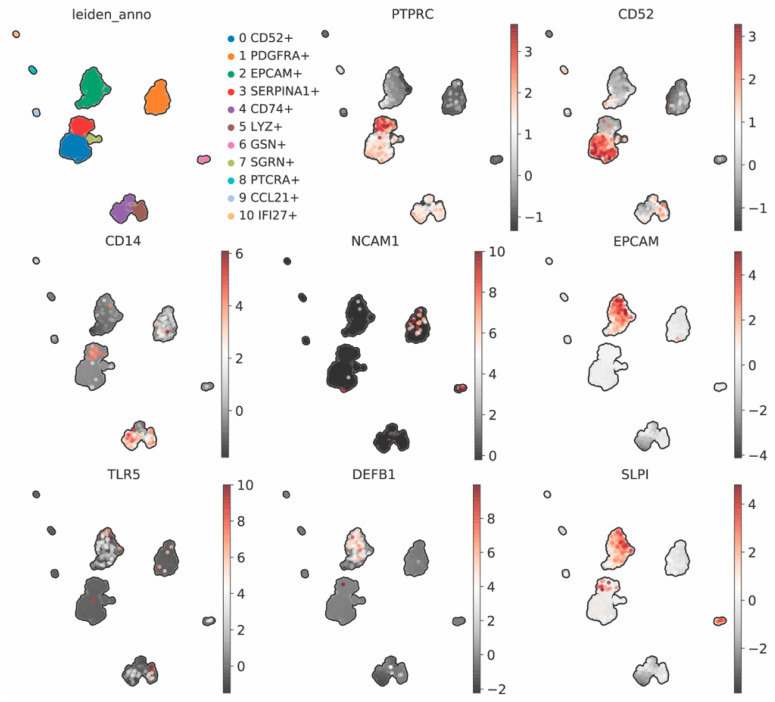
Expression of immune cell markers in the human Fallopian tube using single-cell RNA-sequencing (scRNA-seq) data that was re-analyzed from a previously published dataset [3]. Heterogeneity of cell types in the human Fallopian tube is plotted as Uniform Manifold Approximation and Projection, or UMAP (leiden_anno), T-cells (*CD52*+), leukocytes (*PTPRC*+, also known as *CD45*), monocytes or macrophages (*CD14*+), and NK cells (*NCAM1*+), are compared to the epithelial cell marker, *EPCAM*+. Toll-like receptor 5 (*TLR5*+) and antimicrobial peptide transcripts, such as *DEFB1*+ and *SLPI*+, are detected in the EPCAM+ cell population. Color bars indicate expression levels of each gene in the dataset. Expression of other genes of interest in this dataset is available online: https://www.simplesinglecell.org/winuthayanon/humanFT/ (accessed on 19 January 2022).

**Figure 3 cells-11-01075-f003:**
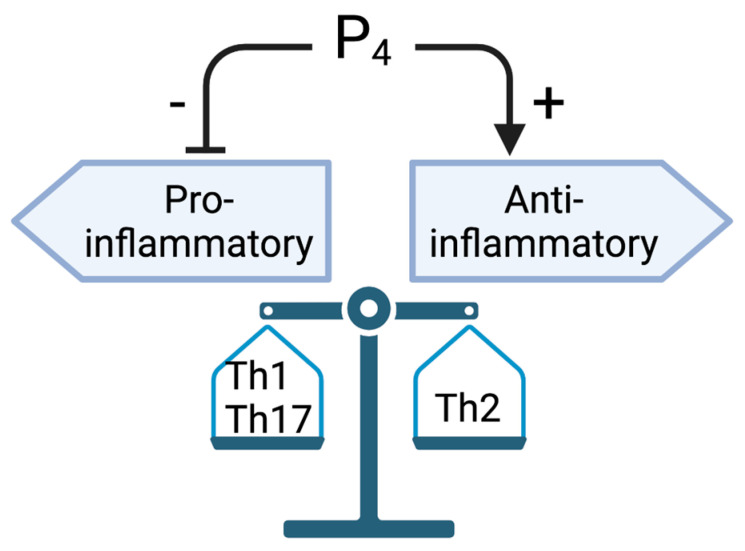
Generalized action of P_4_ on the Th1-, Th2-, and Th17-mediated inflammatory responses. Created with bioRender.com.

**Figure 4 cells-11-01075-f004:**
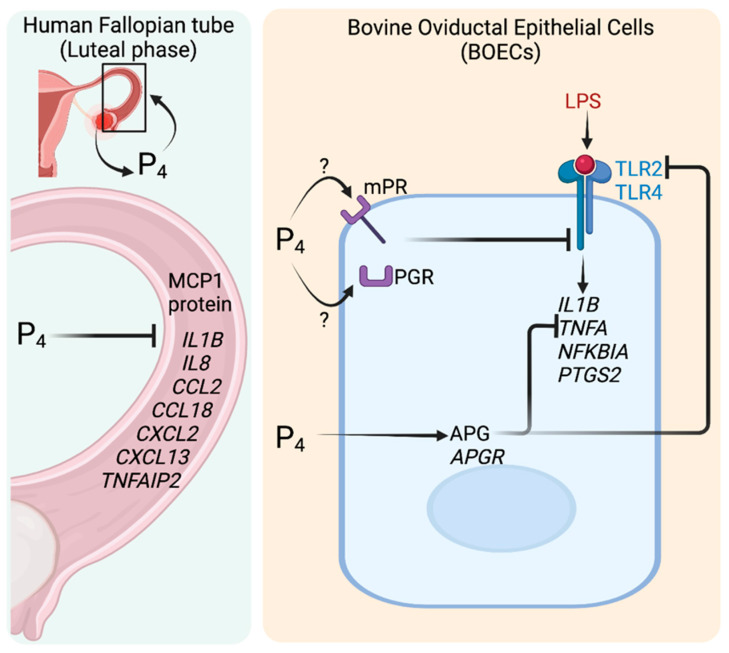
Action of P_4_ on the regulation of cytokine productions (**left**) in human Fallopian tubes during the luteal phase compared to follicular phase and (**right**) in bovine oviductal epithelial cells (BOECs) after LPS stimulation. →, —|, ? signs indicate activation, inhibition, and no information available, respectively. Image created with BioRender.com using information from previously published studies ([79] and [53,80], respectively).

**Table 1 cells-11-01075-t001:** P_4_ regulation of pro- and anti-inflammatory cytokines during pregnancy and in female reproductive tissues. + and − denote pro-inflammatory and anti-inflammatory, respectively. ↑: increased; ↓: decreased; ⊗: inhibited; ⇑: upregulated; ⇓: downregulated.

Cytokine	Pro (+) vs. Anti (−) Inflammatory	Response to P_4_	References
IL1α	+	↓	[63]
IL1β	+	⊗, ↓	[12,64]
IL3	+	↑	[61]
IL4	−	↑	[61,65]
IL6	+ and −	⇑	[66,67,68]
IL7	+	↓	[12]
IL8	+	⊗, ↑	[69,70]
IL10	−	↑	[12,61]
IL11	−	↑	[71]
IL12	+	⊗	[72]
IL13	−	↑	[65]
IL15	+	↑	[73]
IL17	+	⇓	[66]
IL18	+	↓	[74]
IL23	+	⇓	[66]
TNFα	+	↓	[53]
TNFγ	+	↑	[67,68]
IFNγ	+	↓	[12]
LIF ^1^	−	⇑ ^2^	[75,76]

^1^ Leukemia Inhibitory Factor; ^2^ in the presence of estrogen.

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
