# Peer review of "Progesterone and Inflammatory Response in the Oviduct during Physiological and Pathological Conditions"

_cells, 2022, doi:10.3390/cells11071075_

Round 1

Reviewer 1 Report

This is a well-written and interesting review on progesterone signaling in ovidcuts. The authors explored the most important consequences of P4 imbalance on oviductal functional characteristics. They also  highlighted  the relationship between P4, inflammation and the occurrence of  pathological conditions, as ectopic pregnancy, that can dramatically affect woman's health and pregnancy outcome.  I only suggested the authors to shorten sections 4.1 and 4.2 because redundant. They could add some key information in the legend of figure 1.  

Author Response

Reviewer 1

This is a well-written and interesting review on progesterone signaling in oviducts. The authors explored the most important consequences of P4 imbalance on oviductal functional characteristics. They also highlighted the relationship between P4, inflammation and the occurrence of pathological conditions, as ectopic pregnancy, that can dramatically affect woman's health and pregnancy outcome.  I only suggested the authors to shorten sections 4.1 and 4.2 because redundant. They could add some key information in the legend of figure 1.  

Response: The authors thank the reviewer for your time and effort to review our manuscript. To be concise, we have now revised sections 4.1 and 4.2 by removing redundant texts and placed them as a figure legend as suggested.

Reviewer 2 Report

The manuscript entitled “Role of Progesterone on the Inflammatory Response in the Oviduct and Its Impact on Pregnancy Outcomes” is a well-written and informative compilation of the current literature on a yet understudied topic. I would have only a few comments. Most of them are rather suggestions that I kindly ask the authors to consider and implement if they deem them useful.

  1. The focus of the review is largely on progesterone impact on immune responses in the oviduct/uterus. Their mutual influence on pregnancy outcomes is only briefly and superficially considered in the part “Progesterone and Oviductal Inflammation in Pathological Conditions”. I therefore suggest either elaborating more on this matter or changing the title.
  2. I miss kind of a summary comparison between the immunological responses to pathogens, sperm and embryo within the oviduct/female reproductive tract. In the review these are all described in separate paragraphs, but in fact they have to be more or less simultaneously functional. How does the immunological cascade proceed within the tissue (recognition, recruitment etc.; if not known in the oviduct, then deduced from uterus) in the different cases, how is the immunological niche established and what is the role of maternal P4 in this process in distinction to other (e.g. embryonic) factors?
  3. In my point of view part 4 is rather uninformative because it is very general and lacks depth. Therefore, it might be integrated into part 5 and directly linked to the situation in the oviduct / female reproductive tract.
  4. L297 prevents cytokine produced by Th1-lymphocytes … should probably read “cytokine production”?
  5. L 403 (connected to comment 2) The authors cite articles concerning immune responses in other parts of the female reproductive tract (uterus) in the “7.1.2. Sperm” section. Therefore, to me it is not understandable why they do not elaborate at least to some extend on the development of the immunological niche of the embryo within the uterus under “7.1.3. Embryos”.
  6. L 420 (also connected to comment 2) Progesterone and Oviductal Inflammation in Pathological Conditions
    In this part I miss some focus. While in the Chlamydia part the authors explain the involvement of immune competent cells / the inflammatory response of the oviduct or uterus and how these are /might be affected by progesterone (focus of the review), this focus gets lost in the Hydrosalpinx and Ectopic Pregnancy part.
  7. LL422, 424 sexually transmitted infection (STI); STD, please introduce abbreviations

Author Response

Reviewer 2

The manuscript entitled “Role of Progesterone on the Inflammatory Response in the Oviduct and Its Impact on Pregnancy Outcomes” is a well-written and informative compilation of the current literature on a yet understudied topic. I would have only a few comments. Most of them are rather suggestions that I kindly ask the authors to consider and implement if they deem them useful.

Response: The authors thank the reviewer for their valuable insight to help improve the quality of the manuscript. We have now incorporated the comments/suggestions in the revised review as indicated below.

  1. The focus of the review is largely on progesterone impact on immune responses in the oviduct/uterus. Their mutual influence on pregnancy outcomes is only briefly and superficially considered in the part “Progesterone and Oviductal Inflammation in Pathological Conditions”. I therefore suggest either elaborating more on this matter or changing the title.

Response: Thank you for this great suggestion. We agreed with the reviewer that there is not much information in the review regarding the “pregnancy outcomes” aspect due to a lack of knowledge in the literature in this area. Therefore, we have now changed the title of the review to “Progesterone and Inflammatory Response in the Oviduct During Physiological and Pathological Conditions”.

  1. I miss kind of a summary comparison between the immunological responses to pathogens, sperm and embryo within the oviduct/female reproductive tract. In the review these are all described in separate paragraphs, but in fact they have to be more or less simultaneously functional. How does the immunological cascade proceed within the tissue (recognition, recruitment etc.; if not known in the oviduct, then deduced from uterus) in the different cases, how is the immunological niche established and what is the role of maternal P4 in this process in distinction to other (e.g. embryonic) factors?

Response: We agreed with the reviewer. To better assess immunological responses to pathogens, sperm, and embryo within the oviduct, we need to provide a conclusion based on the current findings. Therefore, we have now included a summary comparison at the end of this section, which now reads “In summary, P4 both suppresses pro-inflammatory and stimulates anti-inflammatory responses in the oviduct. Therefore, we can speculate that the presence of P4 from the follicular fluid in the oviduct prevents excessive pro-inflammatory cytokine production from the oviductal epithelial cells. This reduction of pro-inflammatory responses allows the oviductal environment to tolerate the presence of foreign bodies such as sperm and seminal fluid and promote sperm viability in the upper reproductive tract. This tolerative immune response to the sperm could also prime the oviduct to allow for the newly fertilized embryos to develop. Evidence suggests that optimal E2 signaling in the oviduct ensures proper inflammatory balances while embryos are present. Therefore, it is possible that P4 from the functional CL during the preimplantation embryo developmental stage could also antagonize E2-induced pro-inflammation while promoting anti-inflammatory responses to allow for embryo survival in the oviduct”.

  1. In my point of view part 4 is rather uninformative because it is very general and lacks depth. Therefore, it might be integrated into part 5 and directly linked to the situation in the oviduct / female reproductive tract.

Response: Thank you for this suggestion. To orient the general readers that are unfamiliar with the immunology, we intentionally provide general information without going into detail on this “Immunomodulatory Responses and Regulatory Pathways” section before describing the situation in the oviduct or the reproductive tract. Therefore, the authors respectfully request to keep this section. However, we have justified the need at the beginning of this section, which now reads “As there are multiple players and regulatory pathways in the immune response, we have included a brief overview of innate and adaptive immunity (Figure 1) before describing their actions in the oviduct and the reproductive tract”. We have also shortened Sections 4.1 and 4.2 to avoid redundancy by removing similar information from the text and use them as a figure legend for Figure 1.

  1. L297 prevents cytokine produced by Th1-lymphocytes … should probably read “cytokine production”?

Response: Corrected to “cytokine production” as suggested.

  1. L 403 (connected to comment 2) The authors cite articles concerning immune responses in other parts of the female reproductive tract (uterus) in the “7.1.2. Sperm” section. Therefore, to me it is not understandable why they do not elaborate at least to some extend on the development of the immunological niche of the embryo within the uterus under “7.1.3. Embryos”.

Response: The authors thank Reviewer 2 for the comment. This suggestion is also in-line with Reviewer 3’s suggestion to maintain the focus in the oviduct on this section. Therefore, we have now removed the description on the immune responses in the uterus in “7.1.2 Sperm” to emphasize the findings on the oviduct when information is available.

  1. L 420 (also connected to comment 2) Progesterone and Oviductal Inflammation in Pathological Conditions. In this part I miss some focus. While in the Chlamydia part the authors explain the involvement of immune competent cells / the inflammatory response of the oviduct or uterus and how these are /might be affected by progesterone (focus of the review), this focus gets lost in the Hydrosalpinx and Ectopic Pregnancy part.

Response: Reviewer 3 has also suggested similarly. Therefore, we have removed other information that does not directly involve P4/PGR actions within the Hydrosalpinx and Ectopic Pregnancy sections. The removal of these unnecessary contents has made these two sections more concise and focused on the P4 signaling. Thank you for making such important suggestions.

  1. LL422, 424 sexually transmitted infection (STI); STD, please introduce abbreviations

Response: abbreviation added as suggested

Reviewer 3 Report

The manuscript entitled “Role of Progesterone on the Inflammatory Response in the Oviduct and Its Impact on Pregnancy Outcomes” reviewed the role of P4 modulating inflammatory response. The manuscript is well written and presents a large amount of information important to the researchers in this area of study. I feel that this manuscript it will be well accepted and cited by the community in general. I have a couple suggestion to the authors.

Suggestions:

Topic 7.1.2. The connection between P4/Sperm/Inflammatory response within the oviduct is not clear. I would suggest connecting the information or remove this section. The connections presented In the 7.1.3. section is perfect.

Topic 8.2. The connection between oviduct/hydrosalpinx/P4 is weak.  I would suggest improving the connection within this topic or remove this section. The information within the last paragraph is well connected.

Topic 8.3. The connection between oviduct/ectopic pregnancy/P4 is weak.  I would suggest improving the connection within this topic or remove this section. The information within the last two paragraph is well connected.

Author Response

Reviewer 3

The manuscript entitled “Role of Progesterone on the Inflammatory Response in the Oviduct and Its Impact on Pregnancy Outcomes” reviewed the role of P4 modulating inflammatory response. The manuscript is well written and presents a large amount of information important to the researchers in this area of study. I feel that this manuscript it will be well accepted and cited by the community in general. I have a couple suggestion to the authors.

Response: The authors appreciate the reviewer’s comments and their time for reviewing our manuscript. These suggestions have significantly improved the quality of the revised manuscript. We have incorporated the suggestions in the revised review as indicated below.

Suggestions:

Topic 7.1.2. The connection between P4/Sperm/Inflammatory response within the oviduct is not clear. I would suggest connecting the information or remove this section. The connections presented In the 7.1.3. section is perfect.

Response: This suggestion is in-line with the Reviewer 2’s comment. We have now solely focused on the impact of sperm in the inflammatory response in the oviduct and removed information regarding the uterus from the revised version in this “7.1.2 Sperm” section. This information on the sperm and the immunotolerance of the oviduct may provide crucial clues on embryo survival in the oviduct. Therefore, we have added the connection statement to summarize the P4/sperm/embryo/inflammatory response in the oviduct at the end of section 7, which now reads “In summary, P4 both suppresses pro-inflammatory and stimulates anti-inflammatory responses in the oviduct. Therefore, we can speculate that the presence of P4 from the follicular fluid in the oviduct prevents excessive pro-inflammatory cytokine production from the oviductal epithelial cells. This reduction of pro-inflammatory responses allows the oviductal environment to tolerate the presence of foreign bodies such as sperm and seminal fluid and promote sperm viability in the upper reproductive tract. This tolerative immune response to the sperm could also prime the oviduct to allow for the newly fertilized embryos to develop. Evidence suggests that optimal E2 signaling in the oviduct ensures proper inflammatory balances while embryos are present. Therefore, it is possible that P4 from the functional CL during the preimplantation embryo developmental stage could also antagonize E2-induced pro-inflammation while promoting anti-inflammatory responses to allow for embryo survival in the oviduct”.

Topic 8.2. The connection between oviduct/hydrosalpinx/P4 is weak.  I would suggest improving the connection within this topic or remove this section. The information within the last paragraph is well connected.

Response: Reviewer 2 has also suggested similarly. Therefore, we have removed other information that does not directly involve P4/PGR actions within the Hydrosalpinx section and kept only information in the last paragraph as it directly involved P4 action. Thank you for making the suggestion, this section now reads more concisely.

Topic 8.3. The connection between oviduct/ectopic pregnancy/P4 is weak.  I would suggest improving the connection within this topic or remove this section. The information within the last two paragraph is well connected.

Response: Similarly, we have now removed unnecessary information that does not directly involve P4/PGR actions from the Ectopic Pregnancy section. We have retained the information from the last two paragraphs that directly involved P4 action as suggested. Again, thank you for making these suggestions. We greatly appreciate your input.